# Escaping the Subspace Trap: The Role of Optimizer Geometry in Model Width Expansion

Jiabei Chen [* 1 2 3]   Haoyu Wang [* 2 4]   Yang Yu [2]   Yao Xu [1 3]   Liangdong Wang [2]   Guang Liu [2]   Shizhu He [1 3]
Jun Zhao [1 3]   Kang Liu [1 3]

## Abstract

Pre-training large language models from scratch is prohibitively expensive as model scales increase. A practical alternative is Model Width Expansion (MWE), which grows a larger model from a well-pretrained "seed" model to inherit existing capabilities at initialization. However, we identify a phenomenon termed the ***Subspace Trap***: during continual pre-training, parameter updates largely stagnate within a low-dimensional subspace aligned with the initialization, limiting the effective capacity of the expanded model. Our theoretical analysis investigates this issue by attributing it to the function-preserving properties of width expansion. In particular, element-wise adaptive optimizers remain confined to the trap, whereas optimizers that yield an isotropic geometry of parameter updates can escape. To demonstrate the impact of the subspace trap on model performance, we conduct empirical experiments across different model sizes and model families, which show that escaping the trap is principally effective in improving training efficiency and overall model performance. Detailed mechanistic analyses further confirm that escaping the trap indeed activates the new dimensions to encode general knowledge. Our code is available at https://github.com/A-Polar Bear/Model-Width-Expansion.

*Equal contribution [1]The Key Laboratory of Cognition and Decision Intelligence for Complex Systems, Institute of Automation, Chinese Academy of Sciences, Beijing, China [2]Beijing Academy of Artificial Intelligence, Beijing, China [3]School of Artificial Intelligence, University of Chinese Academy of Sciences, Beijing, China [4]Gaoling School of Artificial Intelligence, Renmin University of China, Beijing, China. Correspondence to: Guang Liu <liuguang@baai.ac.cn>, Kang Liu <kliu@nlpr.ia.ac.cn>.

*Proceedings of the 43rd International Conference on Machine Learning*, Seoul, South Korea. PMLR 306, 2026. Copyright 2026 by the author(s).

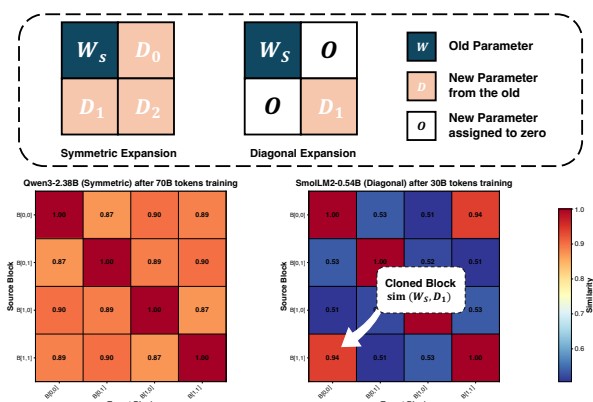

*Figure 1.* **Visualizing Subspace Trap via Block-wise Similarity.** For width expansion factor $k = 2$, we partition $W_L$ into four sub-blocks $B[i, j]$ by splitting both output and input dimensions into two equal groups $(i, j \in \{0, 1\})$. We then compute the pairwise similarity between vectorized blocks between $\text{vec}(B[i, j])$ and $\text{vec}(B[i', j'])$, forming a $4 \times 4$ similarity matrix. Results are shown that even after extensive training, the clone blocks remain highly similar to the seed blocks, evidencing the Subspace Trap.

## 1. Introduction

Empirical scaling laws have served as the guiding principle for the rapid evolution of Large Language Models (LLMs) (Brown et al., 2020; Yang et al., 2025a), establishing a positive correlation between model performance and the scale of parameters, data, and compute (Samragh et al., 2024; Kaplan et al., 2020). While the power laws offer a clear roadmap for capability improvement, they impose a prohibitive computation burden: training frontier-scale models requires hundreds of millions of computational budgets (Grattafiori et al., 2024; Sevilla & Roldán, 2024), rendering the from-scratch training increasingly expensive and environmentally unsustainable (Singh et al., 2025).

Given these challenges, a growing body of work investigates how to leverage the prior knowledge of smaller seed models to achieve more efficient scaling. Model expansion (also known as model growth) has emerged as a distinct and compelling research direction, expanding well-trained base models to larger target sizes (Chen et al., 2015; 2022; Wang et al., 2024b). In contrast to depth expansion strate-

gies that add additional layers (Kim et al., 2024; Wu et al., 2024; Yang et al., 2025b), Model Width Expansion (MWE) enlarges internal dimensions while preserving the number of layers, thereby increasing representational capacity without adding sequential operations to the forward pass. Since autoregressive inference latency is dominated by sequential depth rather than per-step FLOPs, wider matrices are efficiently parallelized on modern GPUs/TPUs with minimal latency overhead (Gesmundo & Maile, 2023; Samragh et al., 2024; Fu et al., 2025). Specifically, MWE methods implement neuron splitting (Chen et al., 2015; 2022; Wang et al., 2024b) or parameter copying (Samragh et al., 2024; Han et al., 2025) to guarantee identical predictions with the seed model at initialization, inducing strong symmetry between newly added parameters and the original ones.

Despite the preliminary success of width expansion in accelerating model training, our empirical observations reveal a persistent phenomenon termed the *Subspace Trap*. As depicted in Figure 1, even after extensive continual pretraining, newly introduced parameters largely stagnate to the original, exhibiting alarmingly high cosine similarities and contributing limited effective capacity. Although random noise is often injected to encourage symmetry breaking, such perturbations are insufficient because the gradient signal is dominated by the well-optimized seed features. Through the lens of loss landscape, our analysis fundamentally attributes this phenomenon to the function-preserving properties of MWE. We prove that function-preserving MWE methods inherit the "River Valley" landscape (Wen et al., 2025), thus mitigating the impact of noise perturbation and leading to symmetric gradients.

Furthermore, we investigate the impact of optimizers on the parameter update trajectory near the subspace trap, especially the ability to escape this confinement. We verify that popular element-wise adaptive optimizer AdamW (Loshchilov & Hutter, 2019), implicitly preserve the initialization-induced geometry and produce updates that remain aligned with the seed subspace, making them ill-suited for escaping the trap. In contrast, optimizers that yield near-isotropic parameter updates in the expanded space (e.g., Muon (Jordan et al., 2024)) provide a principled mechanism for breaking this confinement, enabling effective exploration of newly introduced dimensions. Through an examination of parameter updates, we show that optimizers capable of producing near-isotropic updates actively redistribute optimization energy into previously dormant dimensions, leading to sustained exploration beyond the seed-induced subspace. This shift is accompanied by rapid parameter differentiation and a substantial increase in effective capacity usage.

To validate the impact of the subspace trap and the role of optimizer choice in escaping it, we conduct extensive empirical studies across different model sizes and model families.

Our observations demonstrate that escaping the subspace trap is principally effective in improving both training efficiency and overall model performance. Noticeable, given that the Muon optimizer inherently yields performance gains regardless of the MWE scenario, the comparison further solidifies its advantages within the context of escaping the subspace trap. Beyond end-to-end metrics, a detailed mechanistic analysis reveals that the newly utilized dimensions significantly contribute to the general knowledge learning, further connecting subspace trap with model performance.

## 2. Related Work and Formulation

### 2.1. Related Work

Model growth scales LLMs by initializing a larger model from a smaller pre-trained seed, avoiding training from scratch. Existing growth strategies can be generally categorized into two primary types: depth expansion and width expansion. For depth expansion, earlier approaches often rely on heuristic rules to interpolate and stack layers. For instance, SOLAR (Kim et al., 2024) initializes a deeper model by duplicating the first and last 24 layers of a 32-layer precursor, while LLaMA Pro (Wu et al., 2024) selectively copies only a few specific layers to extend depth. Beyond simple duplication, LESA (Yang et al., 2025b) introduces a learnable method for depth scaling, employing neural networks to predict parameters for new layers based on inter-layer continuity patterns.

Parallel to depth, width expansion strategies focus on enlarging internal dimensions. Net2Net (Chen et al., 2015) was the first work to propose the concept of function-preserving transformations, expanding width via neuron splitting and depth via identity layers to ensure the target model mimics the base at initialization. These principles were subsequently extended to Transformer architectures by bert2BERT (Chen et al., 2022). To achieve lossless expansion in diverse scenarios, LOIRE (Han et al., 2025) proposes a comprehensive framework ensuring strict Function-preserving across all dimensions (width, depth, attention heads) for lifelong learning. Addressing the numerical symmetry in neuron splitting, LEMON (Wang et al., 2024b) introduces weighted initialization coefficients, while LiGO (Wang et al., 2023) adopts a data-driven approach, learning a linear mapping to initialize larger models efficiently.

While foundational works primarily targeted Encoder or Vision architectures, recent advancements have transitioned to the Decoder-only Transformers that dominate the modern LLM landscape. HyperCloning (Samragh et al., 2024) focuses on efficient parameter duplication for LLMs to minimize initialization costs. However, the optimization dynamics of fully trainable, function-preserving width expansion in decoder-only architectures remain poorly understood. We

overcome this gap by diagnosing the geometric pathology underlying such expansions and showing it can be eliminated through optimizer-induced update geometry, without auxiliary training stages or plasticity-limiting constraints.

## 2.2. Formulation of Width Expansion

Let the original weight matrix be $\boldsymbol{W}_S \in \mathbb{R}^{m \times n}$. We define width expansion as an affine transformation that maps from the source space to the target space $\mathbb{R}^{M \times N}$. The expanded weight $\boldsymbol{W}_L$ can be formalized as a linear combination of a Kronecker product term and an additive noise term:

$$\boldsymbol{W}_L = \boldsymbol{P} \otimes \boldsymbol{W}_S + \boldsymbol{\epsilon} \tag{1}$$

where $\boldsymbol{P}$ determines the topological geometry of the expansion, and $\boldsymbol{\epsilon}$ is random noise with proper scale. The noise term $\boldsymbol{\epsilon}$ is not arbitrary; we adopt the counter-balancing noise structure from (Samragh et al., 2024) to ensure strictly lossless function preservation. Detailed formulations are provided in Appendix B.1.

Function-preserving refers to maintaining the original output when expanding the model's weights and input dimensions. It ensures that initialization based on the small model's weight $\boldsymbol{W}_S$ can provide partially learned features, thereby accelerating the grokking phenomenon. Formally, let $\pi_Y$ denote the projection function from new to old, a function-preserving initialization (on module $\mathcal{F}$) satisfies

$$\pi_Y \left[ \mathcal{F}(\boldsymbol{X}; \boldsymbol{W}_L) \right] = \mathcal{F}(\boldsymbol{X}; \boldsymbol{W}_S), \quad \forall X \in \mathcal{X}.$$

Prevalent weight expansion methods with the function-preserving property typically adopt: (1) Symmetric (Samragh et al., 2024) or (2) Diagonal strategies (Shen et al., 2022), which (in the doubling width scenario) are respectively formulated as

$$\boldsymbol{P}_1 := \begin{bmatrix} \frac{1}{2} & \frac{1}{2} \\ \frac{1}{2} & \frac{1}{2} \end{bmatrix}, \qquad \boldsymbol{P}_2 := \begin{bmatrix} 1 & 0 \\ 0 & 1 \end{bmatrix}. \tag{2}$$

## 3. Subspace Trap

In this section, we introduce the *Subspace Trap*, where parameter updates during continual pre-training largely stagnate within a low-dimensional subspace that aligns with the initialization. We begin by providing empirical validation on several model width expansion methods, and then propose a theoretical explanation for how function-preserving properties cause such a problem.

### 3.1. Empirical Observation

**The Expectation.** The additive perturbation $\boldsymbol{\epsilon}$ in Eq. (1) removes exact symmetry at initialization, so in principle gradient-based optimization can update cloned parameters differently. It is therefore natural to hypothesize that the clones specialize and move toward distinct roles in training.

**The Reality.** In practice, however, we observe that this theoretical symmetry breaking is remarkably inefficient. Despite the injection of initialization noise, the cloned parameters under standard optimization (e.g., AdamW) exhibit persistent functional coherence. We visualize this redundancy in Figure 1 by computing the **block-wise similarity** of the trained weights. Specifically, we partition the expanded weight matrix $W_L$ into $k \times k$ functional blocks (where $k$ is the expansion factor) and compute the alignment between the seed blocks and the newly added clone blocks. The resulting heatmaps reveal distinct yet equally problematic patterns depending on the initialization strategy. Symmetric Expansion exhibits global block-wise homogeneity, whereas Diagonal Expansion retains alarmingly high similarity ($> 0.90$) specifically between the seed and clone blocks despite sparse off-diagonals. This persistent redundancy confirms that, regardless of topology, the added parameters fail to diverge, effectively confining the model to the seed's representational rank.

**Comparison with Scratch-Trained Models.** To confirm that the observed redundancy is specific to function-preserving initialization rather than a natural property of trained weight matrices, we compute block similarity for models trained from scratch under identical conditions. As shown in Table 1, scratch-trained models converge to a block similarity of approximately 0.48 regardless of optimizer, far below the $>0.88$ observed in expanded models under AdamW. This confirms that the persistent redundancy in Figure 1 is an artifact of the expansion initialization, not of the architecture or training data.

*Table 1.* **Block Similarity: Expanded vs. Scratch-Trained** (layer 24, up_proj, Qwen3-2.38B). The natural baseline is $\approx 0.48$; expansion pushes similarity above 0.88 under AdamW.

| Setting | Block Similarity |
|---|---|
| Scratch-trained (AdamW) | 0.475 |
| Scratch-trained (Muon) | 0.495 |
| Expanded + AdamW | 0.880 |
| Expanded + Muon | 0.788 |

As further empirical support, Figure 2 shows that the raw gradients at initialization (t = 0) already exhibit a clear spectral cliff around the 50% mark (for 2× expansion), meaning most gradient energy lies in the seed-induced subspace while the newly added dimensions receive negligible signal. This explains why, despite injected noise, cloned parameters still fail to effectively differentiate under standard training.

### 3.2. Theoretical Analysis

We now delve deeper into the fundamental mechanism driving the "Subspace Trap" phenomenon. Our assumptions

regarding training data and model architecture follow the River Valley loss landscape hypothesis (Wen et al., 2025), a framework well-supported by both physical intuition (Liu et al., 2025) and experimental observations (Qiu et al., 2025). We demonstrate that asymmetries introduced by noise perturbations dissipate rapidly within the training gradient flow. Consequently, the parameter gradient matrix becomes confined to a specific subspace, effectively falling into a subspace trap. This structural limitation is one that conventional element-wise optimizers are inherently unable to resolve. The detailed results are provided in Appendix A.

We first formalize the River Valley loss landscape hypothesis as follows. Departing from traditional theoretical analyses, it doesn't rely on specific assumptions on data distribution or model parameterization. Instead, it directly characterizes the loss landscape, enabling a more straightforward analysis of the properties inherent to the gradient matrix. To align with the empirical observations, we extend the original one-dimensional manifold definition to $p$-dimensional.

**Assumption 3.1** ($p$-Dimensional "River"). We assume the existence of a $p$-dimensional "river" $\mathcal{M}$ in the original hypothesis space, any point $w \in \mathcal{M}$ has a gradient $\nabla \mathcal{L}(w_{\mathrm{S}})$ that lies in the subspace spanned by the eigenvectors of the Hessian $\nabla^2 \mathcal{L}(w)$ that correspond to its $k$ smallest eigenvalues, denoted as $\mathrm{span}\{v_i(\nabla^2 \mathcal{L}(w))|i = d - k + 1, \ldots, d.\}$

**Assumption 3.2** (The "Valley" near the "River"). There exists an open set $\mathcal{U}$ containing $\mathcal{M}$ satisfying:

1. $\mathcal{L}(w)$ is analytic with respect to $w$.
2. There exists a constant $\gamma_{\mathrm{max}}$ such that
$$\forall w \in \mathcal{U}, \|\nabla^2 \mathcal{L}(w)\|_{\mathrm{op}} \leq \gamma_{\mathrm{max}}.$$
3. There exist constants $\gamma_{\mathrm{flat}}, \gamma > 0$, such that
$$\forall w \in \mathcal{U}, \lambda_{d-k}(\nabla^2 \mathcal{L}(w)) > \gamma + 4\gamma_{\mathrm{flat}},$$
$$|\lambda_i(\nabla^2 L(w))| < \gamma_{flat}, \forall i \in \{d - k + 1, \ldots, n\}.$$
4. There exist constants $\Delta_{\mathrm{min}}, \Delta, \kappa > 0$ such that
$$\forall w \in \mathcal{U}, \Delta_{\mathrm{min}} < \|\nabla \mathcal{L}(w)\|_2 < \Delta,$$
$$\|\nabla v_i(\nabla^2 \mathcal{L}(w))\|_{\mathrm{op}} < \kappa\gamma/(2\Delta), i \in \{d-k+1, \ldots, n\}.$$
5. For any point $\forall w \in \mathcal{U} - \mathcal{M}$, the gradient $\nabla \mathcal{L}(w)$ is not totally in the subspace $V_{\mathrm{flat}} = \mathrm{span}\{v_{d-k+1}, \ldots, v_d\}$.
6. There exists an open subset $\mathcal{V} \subset \mathcal{U}$ and a constant $r$, $\forall w \in \mathcal{V}$, the $r$-neighbor of the gradient flow starting from $w$ stays in $\mathcal{U}$.

Following the conventions of previous works (Wen et al., 2025), it's reasonable to assume that the parameters of the seed model have already converged to a stable state within the River-Valley loss landscape during training. Building upon this foundation, we formulate the following proposition to establish the structural relationship between the broadened model and the original small one.

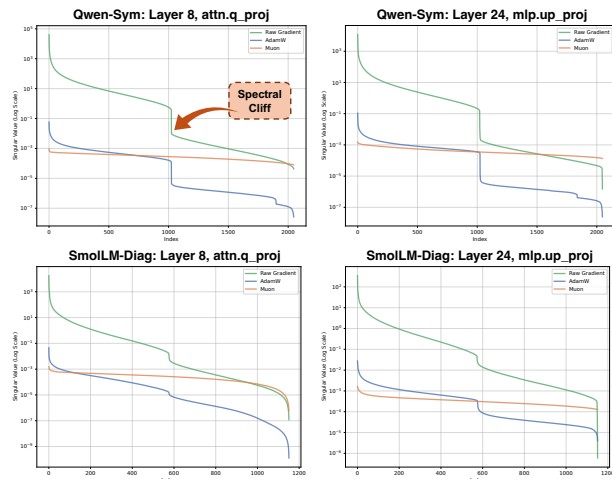

*Figure 2.* **Spectral Diagnosis of Optimizer Updates at Initialization** ($t = 0$). The Raw Gradient exhibits a "spectral cliff" at the 50% mark, reflecting the dominance of the seed subspace. AdamW preserves this bias, confining updates to the seed geometry. In contrast, Muon flattens the spectrum via whitening, enabling isotropic updates that effectively populate the expanded dimensions.

**Proposition 3.3.** *If the original hypothesis space $\mathcal{H}_{\mathrm{S}}$ and the loss function $\mathcal{L}_{\mathrm{S}}$ consist of a river valley loss landscape, then using the weight expansion in Eq. (1), there exists a river valley loss landscape in the new hypothesis space $\mathcal{H}_{\mathrm{L}}$.*

These results demonstrate that, in the absence of noise perturbations, function-preserving expansion schemes confine the model to a river-valley loss landscape. Consequently, the resulting gradient matrix remains restricted to a specific subspace. Since perturbation is introduced, $W_L$ may not lie directly within the "river", but rather in the vicinity of a nearby "valley". The convergence from the valley to the river has already been well-discussed in (Wen et al., 2025). Thus, it's reasonable to disregard the effects of noise and assume that the parameter $w_{\mathrm{L}}$ is already situated within the "river", thereby satisfying Assumption 3.1.

## 4. Influence of Optimizer Geometry

We further apply gradient analysis by providing the following theorem, which directly implies the "Subspace Trap" scheme in the signSGD-style optimizer.

**Theorem 4.1.** *In the river valley landscape of the new hypothesis space, the gradient will still be restricted in the original hypothesis space. With high probability, the signSGD optimizer fails to alter the low-rank nature of the gradient matrix when the SNR is $\Theta(\ln(mn))$.*

Notably, the signSGD is commonly considered as a proxy of AdamW analysis (as a special case of $\beta_1 = \beta_2 = 0$), also implicitly reflecting the limitations of element-wise

optimizers in overcoming the lazy expansion phenomenon.

Given the inherent low-rank property of the gradients, we contend that gradient editing methods specifically targeting matrix rank should be employed. Specifically, an effective remedy must reshape the update spectrum to allocate energy into the remaining directions.

Muon (Jordan et al., 2024) achieves this through Newton-Schulz iterations, which approximate the orthogonal polar factor of the gradient matrix. It has been proven (Chen et al., 2025) that Muon finds solutions satisfying the KKT condition of

$$\min_{W} \mathcal{L}(W), \quad s.t. \quad \|W\|_{\text{op}} \leq \lambda.$$

The spectral norm constraint on the weight matrix prevents any single direction from dominating, which is consistent with the isotropic update geometry: at each step the optimizer allocates comparable magnitude to all nonzero gradient directions, and the converged solution inherits this balanced structure. The subsequent experiments provide empirical evidence to support these assertions.

### 4.1. Spectral Diagnosis of Optimizer Updates

To verify the existence of Subspace Trap, we analyse the singular value spectra of the update matrices $\Delta W$ at the very first optimization step ($t = 0$). Since the "River Valley" structure is a theoretical consequence of the expansion operator in Proposition 3.3, the geometric characteristics of this initial update dictate the subsequent optimization trajectory.

**Element-wise Alignment (AdamW) vs. Orthogonal Intervention (Muon).** Crucially, standard adaptive methods implicitly respect this geometric structure. Lacking a mechanism to rotate the update basis, AdamW preserves the initial spectral bias, confining the trajectory to the seed subspace. This initial confinement establishes the "Subspace Trap". In contrast, Muon fundamentally restructures geometry via spectral whitening. By achieving isotropic energy assignment, it allocates optimization budget to the null space comparable to dominant directions, ensuring the expanded dimensions are populated from the very first step.

### 4.2. Quantifying the Escape from the "River Valley"

While Section 4.1 characterizes the inherent "River Valley" geometry at initialization, we now verify whether the optimizer remains constrained by this structure during the whole training. To quantify this, we adopt an orthogonal decomposition approach consistent with our analysis framework. We define the effective update $\Delta W_{\text{eff}}$ as the orthogonal projection of the total update $\Delta W$ onto the seed-induced subspace $\mathcal{S} = \{P \otimes U \mid U \in \mathbb{R}^{m \times n}\}$. This subspace $\mathcal{S}$ represents the manifold of updates that strictly preserve the

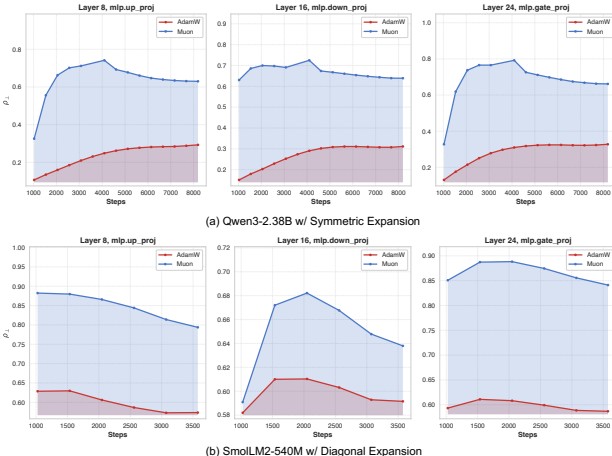

*Figure 3.* **Evolution of Orthogonal Energy Ratio ($\rho_{\perp}$) during training.** The plots demonstrate that while AdamW (Red) remains confined to the low-rank seed subspace, Muon (Blue) rapidly shifts optimization energy toward orthogonal directions, effectively escaping the "Subspace Trap" and utilizing the expanded capacity.

expansion symmetry defined by $P$. Geometrically, $\Delta W_{\text{eff}}$ captures the component of the update that merely replicates the seed model's behavior across the expanded units without introducing functional differentiation.

Accordingly, the total update energy $\|\Delta W\|_F^2$ is analytically decomposed into two orthogonal components:

- **River Subspace Energy ($\mathcal{E}_{\text{river}}$):** The component aligned with the expansion operator $P$. This represents redundant updates that structurally replicate the seed model's behavior, effectively maintaining the parameter trajectory within the subspace ($\|\Delta W_{\text{eff}}\|_F^2$).

- **Orthogonal Subspace Energy ($\mathcal{E}_{\perp}$):** The component lying in the null space of the projection (where the effective functional change is zero, but internal parameters diverge). This represents the symmetry-breaking updates essential for populating the added dimensions and utilizing the model's full theoretical capacity ($\|\Delta W\|_F^2 - \mathcal{E}_{\text{river}}$).

We track the *Orthogonal Energy Ratio*: $\rho_{\perp} = \mathcal{E}_{\perp}/\mathcal{E}_{\text{total}}$ to quantify the extent to which the optimizer breaks free from the Subspace Trap. The results are as follows.

**Persistent Subspace Confinement (AdamW).** As illustrated in Figure 3, AdamW exhibits a persistent adherence to initialization geometry. In Symmetric Expansion, AdamW starts with a negligible orthogonal ratio ($\rho_{\perp} \approx 0.1$) and climbs slowly, only reaching $\approx 0.3$ after thousands of steps. This confirms that element-wise optimization implicitly respects the "Subspace Stability" (Assumption 3.2), treating

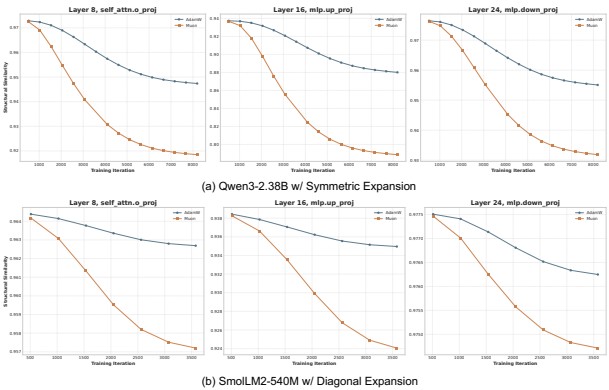

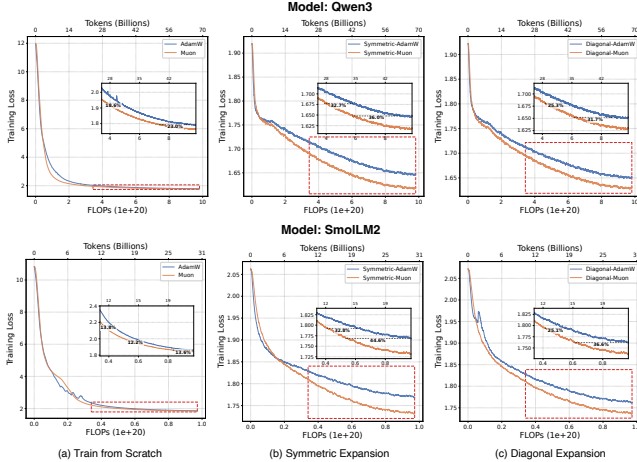

*Figure 4.* **Evolution of Parameter Redundancy.** We track the redundancy $\mathcal{S}_{block}$ during training process. **AdamW (Blue)** remains highly redundant, indicating that gradients are trapped in the subspace of the source model. **Muon (Orange)** rapidly reduces similarity to learn distinct features in the orthogonal space.

the expanded model largely as a redundant container. In Diagonal Expansion, while the baseline starts higher, AdamW often stagnates or even decays, failing to actively populate the off-diagonal blocks required for full-rank interaction.

**Sustained Escape via Isotropy (Muon).** Conversely, Muon's $\rho_{\perp}$ rapidly surges, consistently maintaining a dominant lead over the baseline. For Symmetric Expansion, Muon immediately elevates $\rho_{\perp}$ to the $0.65 - 0.75$ range within the first few hundred steps. For Diagonal Expansion, Muon drives the ratio even higher, approaching $0.90$ in gate projections. This indicates that the vast majority of the optimization budget is actively directed towards the orthogonal directions. This macroscopic shift ensures that the theoretical capacity is translated into optimization energy.

### 4.3. Verifying Parameter Differentiation

While the energy analysis in Section 4.2 demonstrates that Muon generates updates with significant orthogonal components, it remains to be proven that this energy translates into meaningful parameter differentiation rather than random noise. To investigate this microscopic behavior, we track the structural evolution of the expanded weights, specifically measuring the functional coherence between the "seed" parameters and their newly initialized "clones".

We formalize the redundancy metric $\mathcal{S}_{block}$ based on the block-wise similarity of the weight matrix $W_L$. For a width-expanded layer partitioned into a seed block $A$ (top-left) and a clone block $D$ (bottom-right), the similarity is denoted as:

$$\mathcal{S}_{block}(W_L) = \frac{\|A^{\top}D\|_F}{\sqrt{\|A^{\top}A\|_F\|D^{\top}D\|_F}} \tag{3}$$

where $\|\cdot\|_F$ denotes the Frobenius norm. This metric iso-

*Figure 5.* **Impact of Optimizer Geometry on Training Efficiency.** We compare training loss trajectories for **Qwen3-2.38B** (Top Row) and **SmolLM2-540M** (Bottom Row) across three distinct regimes: **(Left)** Training from scratch; **(Center)** Function-Preserving *Symmetric* Expansion; **(Right)** Function-Preserving *Diagonal* Expansion. The insets quantify the **efficiency gain**, defined as the percentage reduction in training tokens required for Muon (Orange) to match the final converged loss of AdamW (Blue).

lates the structural correlation independent of scale; a value near 1.0 signifies a failure to break symmetry, whereas a decreasing value signals the specialization of new parameters.

The empirical trajectories in Figure 4 reveal a distinct contrast in optimization dynamics. Under AdamW, structural similarity remains persistently high, indicating that expanded parameters remain tied to the seed subspace as redundant copies. Conversely, Muon drives a rapid separation of parameter blocks, with similarity dropping significantly early in training. This difference provides microscopic evidence of escaping the "Subspace Trap": Muon's isotropic updates break symmetry and convert added width into utilized capacity, rather than leaving it confined to the seed's low-dimensional features.

## 5. From Parameter Updates to Performance

Former sections show that width expansion can fall into a Subspace Trap: under AdamW, updates stay aligned with the seed subspace, and cloned parameters remain highly redundant. In contrast, Muon produces near-isotropic updates that quickly inject energy into orthogonal directions and drive parameter differentiation. Having established that optimizer geometry can force parameter differentiation, a critical question remains: Does this geometric escape translate into constructive representation learning, or does it merely introduce destructive noise? Breaking the subspace trap is only valuable if the newly activated dimensions encode useful information that accelerates convergence and improves reasoning.

*Table 2.* **Downstream Capability Comparison including Source Baselines.** We compare width-expanded models (SmolLM2-540M and Qwen3-2.38B) against source seed models across diverse benchmarks. While all expansion methods outperform the source baselines, Muon-optimized models consistently improves on average accuracy compared to AdamW, regardless of the initialization strategy. Columns labeled Δ denote the performance gain of Muon over AdamW, with gains highlighted in green.

| Benchmark | SmolLM2 540M | | | | | | | Qwen3 2.38B | | | | | | |
| | Source | Sym. Init | | | Diag. Init | | | Source | Sym. Init | | | Diag. Init | | |
| | (135M) | AdamW | **Muon** | Δ | AdamW | **Muon** | Δ | (0.6B) | AdamW | **Muon** | Δ | AdamW | **Muon** | Δ |
|---|---|---|---|---|---|---|---|---|---|---|---|---|---|---|
| *General Knowledge* | | | | | | | | | | | | | | |
| MMLU | 25.11 | 25.56 | **26.71** | **+1.15** | **26.80** | 26.54 | -0.26 | 52.37 | 52.38 | **53.13** | **+0.75** | 51.37 | **52.34** | **+0.97** |
| MMLU-Pro | 6.88 | 6.51 | **7.70** | **+1.19** | **7.64** | 7.38 | -0.26 | 24.49 | 26.33 | **27.69** | **+1.36** | 26.43 | **26.59** | **+0.16** |
| AGIEval-en | 17.73 | 17.94 | **18.43** | **+0.49** | 18.80 | **18.82** | **+0.02** | 25.83 | **26.30** | 26.27 | -0.03 | **25.47** | 25.03 | -0.44 |
| *Reasoning & Logic* | | | | | | | | | | | | | | |
| Big Bench Hard | 21.55 | 23.65 | **24.39** | **+0.74** | 21.23 | **22.90** | **+1.67** | 35.60 | 41.05 | **44.43** | **+3.38** | 40.15 | **41.90** | **+1.75** |
| ARC Challenge | 29.61 | 32.00 | **33.02** | **+1.02** | 32.00 | 31.14 | -0.86 | 38.31 | 43.69 | **44.80** | **+1.11** | 43.26 | **44.54** | **+1.28** |
| *Commonsense Understanding* | | | | | | | | | | | | | | |
| HellaSwag | 43.34 | 44.47 | **45.81** | **+1.34** | 44.67 | **45.61** | **+0.94** | 53.12 | 59.24 | **61.59** | **+2.35** | 59.05 | **61.23** | **+2.18** |
| Winogrande | 53.59 | 53.28 | **53.67** | **+0.39** | **54.38** | 54.22 | -0.16 | 54.50 | 60.38 | **61.09** | **+0.71** | 58.88 | **61.25** | **+2.37** |
| PIQA | 67.52 | 69.26 | **69.75** | **+0.49** | 69.15 | **69.64** | **+0.49** | 71.00 | 73.94 | **74.10** | **+0.16** | 73.39 | **74.48** | **+1.09** |
| OpenBookQA | 31.80 | 31.00 | **32.40** | **+1.40** | 32.60 | **33.00** | **+0.40** | 35.00 | 36.80 | **37.80** | **+1.00** | 35.60 | **37.00** | **+1.40** |
| **Average Score** | 30.73 | 33.74 | **34.65** | **+0.91** | 34.14 | **34.36** | **+0.22** | 41.72 | 46.67 | **47.88** | **+1.21** | 45.96 | **47.15** | **+1.19** |

We now test whether this geometric escape improves (i) training efficiency (Section 5.2) and (ii) downstream performance (Section 5.3). To separate expansion-specific gains from Muon's general optimization advantage, we also include train-from-scratch baselines with the same target architectures. Finally, we use SVD pruning to verify that the newly activated directions store functional information rather than noise.

## 5.1. Experimental Setup

**Model Architectures.** To assess the scalability of our approach across different regimes, we employ two distinct open-source model families: SmolLM2 (allal et al., 2025) and Qwen3 (Yang et al., 2025a). We utilize SmolLM2-135M and Qwen3-0.6B as source checkpoints, which are then upscaled via our width expansion operators to produce **SmolLM2-540M** and **Qwen3-2.38B**, respectively. For comprehensive details regarding model specifications, training datasets, and hyperparameters, please refer to Appendix B.

**Evaluations.** The evaluation focuses on their performance across general knowledge, reasoning and commonsense understanding. All benchmarks are evaluated through `lm-eval-harness` (Biderman et al., 2024) framework with settings detailed in Appendix C.1.

## 5.2. Training Efficiency Improvement from Geometry

A potential confounder in analysing expansion dynamics is the intrinsic superiority of the optimizer; if Muon were universally faster than AdamW regardless of geometry, the gains observed in expansion would be trivial attribution

errors. To isolate the benefit specific to breaking the "Subspace Trap" we compare the relative speedup in Train-from-Scratch baselines versus Function-Preserving Expansion settings. We quantify this efficiency gain as the percentage reduction in training tokens needed to match the converged loss of the AdamW baseline.

We first establish a baseline for general optimization advantage by examining the standard train-from-scratch regime. As illustrated in Figure 5 (Left), Muon demonstrates a moderate yet bounded advantage over AdamW, achieving speedups of approximately 23.0% for Qwen-2.38B and 13.6% for SmolLM2-540M. These results quantify a baseline general gain ($\Delta_{scratch}$) of roughly 15%–20%, reflecting Muon's ability to handle standard optimization landscapes more effectively than element-wise methods.

In comparison, the efficiency gap becomes substantial within function-preserving expansion settings, surpassing the established baseline. Under Symmetric Expansion, Muon accelerates training by 32.7%–36.0% for Qwen-2.38B and up to 44.6% for SmolLM2, while Diagonal Expansion exhibits consistent trends with speedups ranging from 25.1% to 36.6%. Crucially, the efficiency gain in the expansion regime is approximately double that of the scratch regime ($\Delta_{exp} \gg \Delta_{scratch}$), yielding a gain of nearly 20%.

This substantial differential suggests that Muon mitigates specific geometric bottlenecks inherent to expansion, rather than functioning merely as a faster optimizer. While AdamW appears constrained by the low-rank initialization—tending to treat the expanded architecture as a redundant extension of the seed solution—Muon's spectral

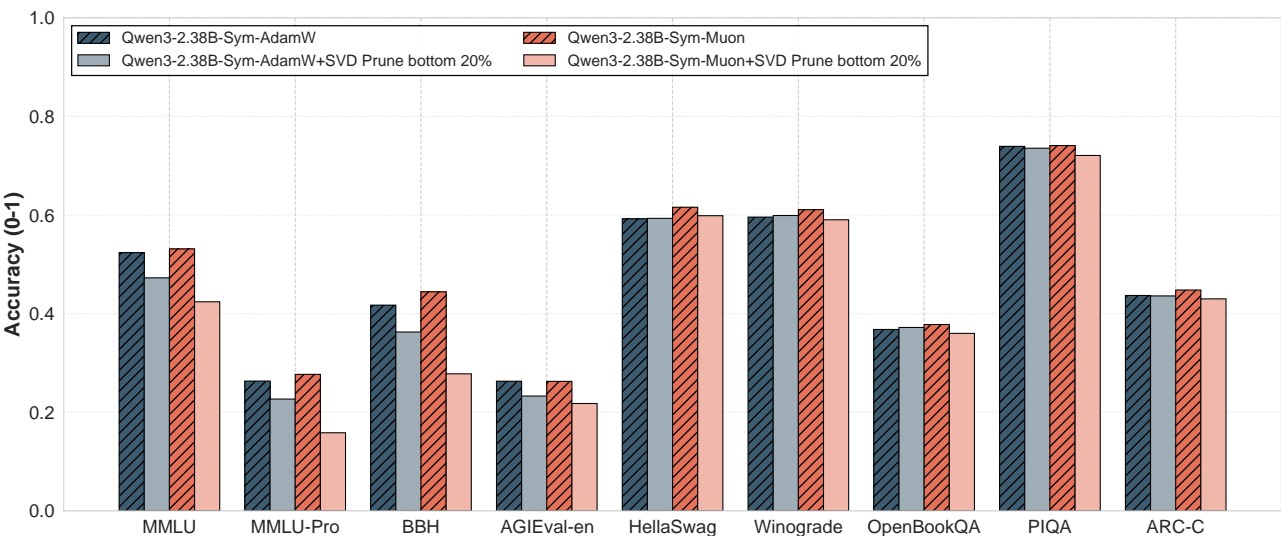

*Figure 6.* **SVD Pruning Sensitivity Analysis.** Performance comparison before and after removing the bottom 20% of singular values. **AdamW (Blue)** shows negligible degradation confirming that the tail components are redundant. Conversely, **Muon (Orange)** suffers a significant performance drop, proving that it stores critical knowledge even in the tail components, thereby achieving higher capacity.

whitening facilitates exploration into the null space. This mechanism enables more efficient utilization of the added parameters compared to element-wise adaptation, effectively leveraging the expanded model's theoretical capacity. From a practical standpoint, this means that width expansion should be paired with geometry-aware optimization; otherwise, a significant fraction of the added parameters remains wasted on redundant updates.

### 5.3. Downstream Capabilities Scaling

We further substantiate the optimization gains observed in Section 5.2 by evaluating whether the accelerated convergence translates into tangible performance gains across downstream benchmarks.

First, comparing the expanded models against their unexpanded seed models confirms the validity of width scaling. As detailed in Table 2, expanded architectures consistently outperform the smaller source baselines. However, the significance of improvement varies by optimizers. While AdamW yields modest gains over the source baseline, Muon achieves a more distinct improvement. On Qwen3-2.38B with Symmetric Expansion, Muon improves the average score by 6.16 points over the 0.6B source, compared to 4.95 for AdamW—indicating that Muon converts a greater fraction of the added parameters into functional capacity. This is consistent with the mechanistic evidence in Sections 4.2–4.3: the orthogonal energy that Muon injects into the expanded dimensions translates into measurable downstream gains, not merely parameter-level divergence.

Moreover, this superiority is robust across diverse evalua-

tion domains. On the Qwen3-2.38B model, Muon achieves a higher Average Score of 47.88 compared to 46.67 for AdamW. This broad consistency suggests that optimizer-level geometry control promotes a globally effective expansion dynamic, mitigating the tendency to collapse into the seed model's subspace.

### 5.4. Proving Capacity Utilization via SVD Pruning

The parameter differentiation observed in Section 4.3 is a necessary condition for effective expansion, but it is not sufficient; random noise can also induce symmetry breaking without necessarily encoding functional structures. To rigorously verify that Muon's updates encode valid knowledge, we conduct a **Spectral Pruning Analysis**. Our hypothesis is grounded in information theory: a model that effectively utilizes its expanded capacity for knowledge storage should be more sensitive to spectral truncation than a structurally redundant model.

**Methodology.** We apply Singular Value Decomposition (SVD) to the weight matrices of the fully trained models. For each layer weight $\mathbf{W}$, we decompose it as $\mathbf{W} = \mathbf{U}\boldsymbol{\Sigma}\mathbf{V}^{\top}$ and explicitly zero out the bottom 20% of the singular values (representing the least significant principal components). We then reconstruct the weights and evaluate the performance on downstream benchmarks.

**Results.** As illustrated in Figure 6, the results confirm our hypothesis with a stark contrast in degradation patterns:

- **AdamW Robustness as an Indicator of Redundancy:** The AdamW-optimized model exhibits remarkable ro-

bustness to spectral truncation, largely maintaining its original performance despite the removal of tail components. While typically a desirable trait, in the context of width expansion, this insensitivity signals structural redundancy. It suggests that the optimization trajectory has remained confined to the principal components inherited from the seed model, rendering the expanded "tail" dimensions functionally inert.

- **Muon Sensitivity as a Sign of Density:** In contrast, the Muon-optimized model demonstrates significant sensitivity to pruning, suffering a pronounced performance decline under the same conditions. This degradation provides evidence that the orthogonal energy identified in Section 4.2 has been utilized to encode critical functional features within the lower-energy subspaces. The expanded dimensions are not populated by arbitrary noise, but by information-bearing components essential for downstream reasoning.

This verification completes our mechanistic investigation. It substantiates that Muon's geometric intervention achieves more than just numerical parameter diversification, ensuring that the computational cost of the larger architecture translates into utilized representational capacity.

## 6. Conclusion

In this work, we identify the "Subspace Trap" as a critical bottleneck in function-preserving model width expansion, where newly added parameters fail to diverge from the seed model's initialization under standard element-wise optimization. By analysing the geometry of the loss landscape, we demonstrate that this stagnation arises from spectral collapse and can be effectively resolved by enforcing isotropic update geometry via spectral whitening. Crucially, this activates the new dimensions to encode valid functional knowledge. Our empirical results across the SmolLM2 and Qwen3 confirm that this geometric intervention not only accelerates training convergence but also significantly enhances downstream capabilities by ensuring the effective utilization of expanded capacity. These findings establish a principled optimization framework for efficient model scaling, offering a sustainable alternative to training large language models from scratch.

**Limitations.** Our experiments focus on decoder-only Transformers (SmolLM2 and Qwen3 families). While the River-Valley loss landscape has been empirically observed across CNNs, ViTs, and encoder-decoder models (Wen et al., 2025), we have not directly verified the Subspace Trap in those architectures. The theoretical analysis relies on the River-Valley landscape assumptions (Assumptions 3.1–3.2), which require an analytic loss function, bounded Hessian, and a clear eigen gap. The specific conditions under which these assumptions break down remain to be characterized.

## Acknowledgements

This work was supported by the Beijing Natural Science Foundation (L243006), the National Natural Science Foundation of China (No.62376270), and the independent research project of the Key Laboratory of Cognition and Decision Intelligence for Complex Systems. We would like to thank the Beijing Academy of Artificial Intelligence (BAAI) for providing an excellent research environment where this project was completed, with special thanks to the OpenSeek and FlagOS teams for their unified AI system software stacks and training frameworks.

## Impact Statement

This paper presents work whose goal is to advance the field of Machine Learning, specifically by enhancing the training efficiency of Large Language Models through geometric optimization control during width expansion. The proposed method aims to resolve the "Subspace Trap" phenomenon, thereby reducing the computational resources required for scaling models and promoting more environmentally sustainable AI development. There are many potential societal consequences of our work, none which we feel must be specifically highlighted here beyond the general implications of advancing the capabilities and accessibility of generative AI.

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

# A. Theoretical Details

## A.1. Loss Landscape Analysis

*Proof of Proposition 3.3.* We abuse the notation to denote the parameters in the original hypothesis space as $w_{\mathrm{S}} = [\mathrm{vec}(W_{\mathrm{S}}^1), \dots, \mathrm{vec}(W_{\mathrm{S}}^n)]^\top$, and the parameters in the new hypothesis space $w_{\mathrm{L}}$ likewise ($n$ is the number of learnable matrices). we highlight that the relation between $w_{\mathrm{S}}$ and $w_{\mathrm{L}}$ is

$$w_{\mathrm{L}} = Jw_{\mathrm{S}}, \quad \nabla\mathcal{L}(w_{\mathrm{L}}) = J\nabla\mathcal{L}(w_{\mathrm{S}}), \quad \nabla^2\mathcal{L}(w_{\mathrm{L}}) = J^\top\nabla^2\mathcal{L}(w_{\mathrm{L}})J, \dots$$

where $J = P \otimes I_{N \times M} \otimes I_n$ is the transformation from original hypothesis space to the new (if all the learnable matrix have the shape of $N \times M$). Since $J$ is a static linear transformation, it guarantees the new loss landscape satisfy Assumption 3.2.1. Without loss of generality, let $\nabla^2\mathcal{S}(w_{\mathrm{L}}) = V_{\mathrm{S}}^\top\Sigma_{\mathrm{S}}V_{\mathrm{S}}$, then

$$\nabla^2\mathcal{L}(w_{\mathrm{L}}) = J^\top(V_{\mathrm{S}}^\top\Sigma_{\mathrm{S}}V_{\mathrm{S}})J = (V_{\mathrm{S}}J)^\top\Sigma_{\mathrm{S}}(V_{\mathrm{S}}J).$$

If Assumption 3.1 is satisfied in the original hypothesis space, we have

$$\nabla\mathcal{L}(w_{\mathrm{S}}) = \sum_{i=d-k+1}^{d} a_i \cdot v_i^{\mathrm{S}} \Rightarrow \nabla\mathcal{L}(w_{\mathrm{L}}) = \sum_{i=d-k+1}^{d} a_i \cdot (v_i^{\mathrm{S}}J) = \sum_{i=d-k+1}^{d} a_i \cdot v_i^{\mathrm{L}},$$

which guarantees that there exists a river in the new hypothesis space. One can prove Assumption 3.2.5 in a similar way.

Also, there exists

$$\gamma'_{\max} = \lambda(P)\gamma_{\max}, \gamma' = \lambda(P)\gamma, \gamma'_{\mathrm{flat}} = \lambda(P)\gamma_{\mathrm{flat}},$$

such that Assumption 3.2.2, 3.2.3 holds[1]. Also, according to the Cauchy-Schwarz inequalities, we have

$$\|\nabla\mathcal{L}(w_{\mathrm{L}})\| = \|P\|_{\mathrm{F}}\|\nabla\mathcal{L}(w_{\mathrm{S}})\| < \|P\|_{\mathrm{F}}\Delta,$$

by setting $\Delta' = \|P\|_{\mathrm{F}}\Delta$ and $\Delta'_{\min} = \Delta_{\min}$ we obtain Assumption 3.2.4. Combining all the results finishes the proof. $\square$

## A.2. Gradient Analysis

Due to the introduction of noise, $W_L$ may not lie directly within the "river" of the new hypothesis space, but rather in the vicinity of a nearby "valley". Regarding the convergence from the valley to the river, Wen et al. (2025) has provided a detailed discussion; here, we present their convergence theorem for gradient flow (1-dimension river) as a preliminary foundation.

**Theorem A.1.** *If a loss $\mathcal{L}$ is a river valley, the gradient flow $w(t)$, will obey the following dynamics:*

1. *The iterate will first converge to a neighborhood of the river. The distance between the iterate and the river is bounded after a constant converge time $T_{converge} = 2\log(2\Delta/(\kappa\Delta_{\min}))/\gamma$.*

$$dist(w(T_{converge}), \mathcal{M}) = \min_t \|x(t) - w(T_{converge})\|_2 \le 2\kappa\Delta/\gamma.$$

2. *After convergence, the iterate will track the river closely with the same pace as the reference flow. There exists a time shift $T_0$ depending on $w(T_{converge})$, such that for any $t \in [T_{converge}, T_{\max}]$ for $T_{\max}$ defined in Assumption 2.6, there exists a $\tilde{t} \in [(1-\epsilon)T, (1+\epsilon)T]$ satisfying that,*

$$\|x(T_0 + \tilde{t}) - w(t)\|_2 \le 2\kappa\Delta/\gamma,$$

*for $\epsilon = 30\kappa$.*

With the guarantees provided by the theorem, we directly disregard the effects of noise and assume that the parameter $w_{\mathrm{L}}$ is already situated within the 'river,' thereby satisfying Assumption 3.1.

---

[1] $\lambda(P) = \{1, 1\}$ or $\{1, 0\}$ in the Eq. (2)

**Lemma A.2.** *In the river valley landscape of the new hypothesis space, the gradient will still be restricted in the original hypothesis space, which is restricted in the original hypothesis space in a projection view.*

*Proof.* We first analyse the properties of the landscape by checking the Hessian matrix:

$$\nabla^2 \mathcal{L}(w_{\mathrm{L}}) = (P^\top P) \otimes (V_{\mathrm{S}}^\top \Sigma_{\mathrm{S}} V_{\mathrm{S}}),$$

whose eigenvectors will be $\{v_i \otimes v_j | v_i \in \mathrm{Eigenvec}(P), v_j \in \mathrm{Eigenvec}(\nabla^2 \mathcal{L}(w_{\mathrm{S}}))\}$ because

$$\begin{aligned}
&\nabla^2 \mathcal{L}(w_{\mathrm{L}})(v_i \otimes v_j) = (P \otimes I)^\top \nabla^2 \mathcal{L}(w_{\mathrm{S}})(P \otimes I)(v_i \otimes v_j) \\
=&(P \otimes I)^\top \nabla^2 \mathcal{L}(w_{\mathrm{S}})(Pv_i \otimes v_j) = (P \otimes I)^\top \nabla^2 \mathcal{L}(w_{\mathrm{S}})(\sigma_i v_i \otimes v_j) \\
=&(P \otimes I)^\top (\sigma_i v_i \otimes \nabla^2 \mathcal{L}(w_{\mathrm{S}}) v_j) = (P \otimes I)^\top (\sigma_i v_i \otimes \sigma_j v_j) \\
=&\sigma_i \sigma_j (P^\top v_i) \otimes v_j = \sigma_i^2 \sigma_j v_i \otimes v_j.
\end{aligned}$$

The matrix defined in Eq. (2) has eigenvectors

$$V_1 = \begin{bmatrix} \frac{1}{\sqrt{2}} & \frac{1}{\sqrt{2}} \\ 0 & 0 \end{bmatrix}, \quad \text{or} \quad V_2 = \begin{bmatrix} 1 & 0 \\ 0 & 1 \end{bmatrix},$$

which means the update will still be restricted in the original hypothesis space (in the sense of symmetry). □

**Lemma A.3.** *Let gradient matrix $G$ is low-ranked. The SignSGD performs $\Delta W \triangleq sign(G + E)$, where $E$ is the estimated noise following $\mathcal{N}(\mathbf{0}, \sigma^2 \mathbf{I})$. Let the minimum signal strength $\mu_{\min}$ of $G$ be $\mu_{\min} \triangleq \min_{i,j} |G_{ij}|$. If the Signal-to-Noise $\mu_{\min}/\sigma$ is $\Theta(\ln(mn/\delta))$, then the probabilities that SignSGD produces low rank update as $G$ is $1 - \delta$.*

*Proof.* For any element $(i, j)$, a sign flip (i.e., $\mathrm{sign}(\hat{G}_{ij}) \neq \mathrm{sign}(G_{ij})$) occurs if and only if the noise $E_{ij}$ exceeds the signal magnitude $|G_{ij}|$ in the opposite direction. By applying the Gaussian tail bound, we have

$$\begin{aligned}
P(\mathrm{sign}(\hat{G}_{ij}) \neq \mathrm{sign}(G_{ij})) &= P(\mathrm{sign}(G_{ij} + E_{ij}) \neq \mathrm{sign}(G_{ij})) \\
&= P(E_{ij} \text{ has opposite sign to } G_{ij} \text{ and } |E_{ij}| > |G_{ij}|) \\
&\leq P(|E_{ij}| > \mu_{\min}) \\
&= 2 \cdot P\left(Z > \frac{\mu_{\min}}{\sigma}\right) \quad (\text{where } Z \sim \mathcal{N}(0,1)) \\
&\leq 2 \exp\left(-\frac{\mu_{\min}^2}{2\sigma^2}\right)
\end{aligned}$$

We aim for the entire matrix's signs to be perfectly recovered with a probability of at least $1 - \delta$. Let $\mathcal{A}$ denote the event that $\mathrm{sign}(\hat{G}) = \mathrm{sign}(G)$ (i.e., no sign flips occur among all $mn$ elements). By applying the Union Bound (Boole's inequality) to the complementary event $\mathcal{A}^c$:

$$P(\mathcal{A}^c) = P(\exists (i,j) \text{ s.t. } \mathrm{sign}(\hat{G}_{ij}) \neq \mathrm{sign}(G_{ij})) \leq \sum_{i,j} P(\mathrm{sign}(\hat{G}_{ij}) \neq \mathrm{sign}(G_{ij})) \leq mn \cdot 2 \exp\left(-\frac{\mu_{\min}^2}{2\sigma^2}\right).$$

To ensure $P(\mathcal{A}) \geq 1 - \delta$, we set the upper bound of the failure probability to be at most $\delta$:

$$2mn \exp\left(-\frac{\mu_{\min}^2}{2\sigma^2}\right) \leq \delta$$

Solving this inequality yields the condition for the critical Signal-to-Noise Ratio (SNR). □

*Proof of Theorem 4.1.* combining Theorem A.1 Lemma A.2 and A.3 yields the results. □

# B. Architectures and Training Details

## B.1. Width Expansion Details

In this section, we detail the specific initialization strategies used to map the parameters from the source model $W_S$ to the expanded target model $W_L$. Following the methodology established in HyperCloning (Samragh et al., 2024), we focus on function-preserving transformations that ensure the larger model inherits the capabilities of the seed model at initialization. We denote the expansion factor as $n$ (where $n = 2$ represents doubling the width).

**Linear Layer Expansion.** For a linear layer performing the operation $y = Wx + b$, we consider the case where both input and output dimensions are expanded.

- **Symmetric Expansion** The Symmetric strategy distributes the weights of the source model across the expanded blocks to maintain the forward pass consistency. This approach ensures that the output of the expanded layer is identical to the cloned output of the source layer ($y_L = [y_S, y_S]^T$). For a 2-fold expansion ($n = 2$), the weight matrix $W_L$ is initialized as a block matrix where each block is a scaled copy of $W_S$:

$$W_L^{Sym} = \begin{bmatrix} \frac{1}{2}W_S & \frac{1}{2}W_S \\ \frac{1}{2}W_S & \frac{1}{2}W_S \end{bmatrix} + \begin{bmatrix} E_1 & -E_1 \\ E_2 & -E_2 \end{bmatrix}$$

  The noise term $\epsilon$ is constructed using random Gaussian matrices $E_1, E_2 \sim \mathcal{N}(0, \sigma^2)$. The scaling factor of $\frac{1}{n}$ (here $\frac{1}{2}$) is applied to normalize the contribution of the expanded input dimension. The bias vector is simply duplicated $n$ times:

$$b_L = \begin{bmatrix} b_S \\ b_S \end{bmatrix}$$

- **Diagonal Expansion** The Diagonal strategy initializes the expanded weight matrix as a block-diagonal matrix. Unlike the symmetric approach, this strategy enforces sparsity at initialization by setting off-diagonal blocks to zero. For a 2-fold expansion, the formulation is:

$$W_L^{Diag} = \begin{bmatrix} W_S & 0 \\ 0 & W_S \end{bmatrix} + \begin{bmatrix} E_1 & -E_1 \\ E_2 & -E_2 \end{bmatrix}$$

  In this setup, the duplicated neurons operate independently at initialization. While this satisfies the function-preserving property, our main text analysis suggests that it faces distinct optimization challenges compared to the symmetric approach.

**Attention and Normalization Layers.** To ensure complete function-preservation across the Transformer block, we adopt the following strategies for other components, referencing the implementation details in HyperCloning.

- **Attention Layers**: When expanding the number of attention heads, we simply duplicate the heads from the source model.

- **Layer Normalization**: The weights ($\gamma$) and biases ($\beta$) of LayerNorm (or RMSNorm) are duplicated $n$ times to match the expanded hidden dimension. This ensures that $l(x_D) = [l(x_S), l(x_S)]^T$. Positional Embeddings: Similar to layer normalization, the positional embedding vectors are repeated $n$ times to match the expanded hidden size.

## B.2. Model Architectures

In this section, we detail the model architectures employed in our experiments. We utilize two distinct model families, SmolLM2 and Qwen3. Table 3 summarizes the architectural specifications for both the Base (seed) models and the corresponding Target (expanded) models.

The Base models serve as the pre-trained initialization for our expansion methods. The Target specifications define the final architecture after width expansion; notably, these specifications are also used to instantiate the "Train from Scratch" baselines referenced in Section 5.2, ensuring a fair comparison between expanded models and standard pre-training.

*Table 3.* Summary of base and target model architectures.

|  | **Model** | **#L** | **#H** | $\mathbf{d_{model}}$ | $\mathbf{d_{FFN}}$ |
|---|---|---|---|---|---|
| **base** | SmolLM2 135M | 30 | 9 | 576 | 1536 |
| **target** | SmolLM2 540M | 30 | 18 | 1152 | 3072 |
| **base** | Qwen3 0.6B | 30 | 16 | 1024 | 3072 |
| **target** | Qwen3 2.38B | 30 | 32 | 2048 | 6144 |

## B.3. Datasets

Across all experiments, we use NVIDIA's Nemotron-CC-v2 dataset (Basant et al., 2025), which consists of curated Common Crawl snapshots and synthetic data totaling up to 6.6T tokens. Due to computational cost, our training runs do not consume the full corpus. To ensure balanced exposure across Nemotron-CC-v2 sub-datasets, we shuffle the data shards prior to training. We use a fixed shuffling seed for all experiments to control for potential effects of data order on the results.

## B.4. Training Details

Across all experiments, we use a micro-batch size of 2 sequences per device and a global batch size of 2048 sequences. We employ a cosine learning-rate schedule with linear warmup for the first 1,000 steps to the peak learning rate, followed by cosine decay to 10% of the peak learning rate. Unless otherwise stated, we use a weight decay of 0.1. For AdamW, we set $\beta_1 = 0.9$ and $\beta_2 = 0.95$; for Muon, we use $\beta = 0.95$. We use FlagScale [2] as our training framework. Our models are trained on 64 GPUs with context size of 4096 and learning rate of 1e-4. To follow the commonly used 20× scaling-law heuristic, we set the training-token budgets to **30B** tokens for SmolLM2-540M and **70B** tokens for Qwen3-2.38B, respectively.

## C. Evaluations

### C.1. Evaluation Settings

**Model Evaluations.** Table 4 details the evaluation settings and frameworks used for our downstream performance evaluations.

*Table 4.* Evaluation settings and benchmark sources for models.

| **Benchmark** | **Metric** | **Settings** |
|---|---|---|
| **General Knowledge** | | |
| MMLU (Hendrycks et al., 2021) | exact match, get response | 5-shot |
| MMLU-Pro (Wang et al., 2024a) | exact match, custom-extract | 5-shot, CoT |
| AGIEval-en (Zhong et al., 2024) | strict match, acc | 0-shot |
| **Reasoning & Logic** | | |
| Big Bench Hard (Suzgun et al., 2023) | exact match, get-answer | 3-shot, CoT |
| ARC Challenge (Clark et al., 2018) | logprobs | 0-shot |
| **Commonsense Understanding** | | |
| HellaSwag (Zellers et al., 2019) | logprobs | 5-shot |
| Winogrande (Sakaguchi et al., 2020) | logprobs | 5-shot |
| PIQA (Bisk et al., 2020) | logprobs | 5-shot |
| OpenBookQA (Mihaylov et al., 2018) | logprobs | 5-shot |

---

[2] https://github.com/flagos-ai/FlagScale

