# OpenReview forum: "Escaping the Subspace Trap: The Role of Optimizer Geometry in Model Width Expansion"
_ICML.cc/2026/Conference — ICML 2026 regular_

### Official Review · Reviewer_v95r · 2026-03-03

**Soundness:** 3
**Presentation:** 3
**Significance:** 3
**Originality:** 3
**Overall Recommendation:** 5
**Confidence:** 5

**Summary:**

This paper analyzes limitations of function-preserving model width expansion for continual pretraining of language models. It finds that common function-preserving expansion initializations can trap optimization in a low-dimensional, seed-induced subspace near the pretrained weight manifold, leaving newly added parameters under-utilized and slowing convergence. The authors mitigate this by changing the optimizer geometry: they use spectral/orthogonalized updates (Muon) rather than coordinate-wise adaptive updates (e.g., AdamW), which helps inject learning signal into the expanded directions and escape the subspace trap. Across extensive experiments, they show this approach reduces redundancy after expansion, accelerates training, and improves downstream performance.

**Compliance With Llm Reviewing Policy:**

Affirmed.

**Final Justification:**

The paper is technically solid and meaningful,. It explains why using pretrained models weights as initialization is not always the best option when pretraining  a model on on a new task.  However, some concerns about the breadth of empirical validation and the limits of the theoretical assumptions remain only partially resolved( The method was  mainly tested on decoder only)

**Key Questions For Authors:**

1. **Generality beyond LLMs:** Do the key findings (subspace trap + optimizer-geometry dependence) also hold for **image classification** settings and non–decoder-only architectures (e.g., CNNs, ViTs, encoder–decoder Transformers)?

2. **Sensitivity to optimization choices:** How do **optimizer hyperparameters** and the **learning-rate schedule** affect training after expansion? In particular, do warm-started expanded models require different tuning (e.g., LR, betas, weight decay, warmup, optimizer-state reset) compared to training from scratch?

3. **When assumptions fail:** The theory relies on assumptions that may not always hold. Are there identifiable **practical regimes** where these assumptions are violated, yet the proposed approach still applies? Conversely, are there known cases where standard expansion succeeds and aligns with (or contradicts) the methodology proposed here?

**Limitations:**

The main limitation i have for this paper is the gap between the scope of its claims and the scope of its empirical validation: it presents the subspace trap as a broadly general phenomenon, but the experiments are largely confined to a single architectural setting **decoder-only language models**. As a result, it remains unclear how well the conclusions transfer to other model families and domains.

**Strengths And Weaknesses:**

## Strengths

- **Soundness:** The paper is technically robust, with claims well supported by both theoretical analysis of the *“River Valley”* loss landscape and extensive empirical evaluations across multiple model families (e.g., SmolLM2 and Qwen3).
- **Presentation:** The narrative is clearly structured and easy to follow. Visual aids such as block-wise similarity heatmaps and spectral cliff diagrams effectively illustrate complex geometric concepts like the *“Subspace Trap.”*
- **Significance:** The work addresses a high-impact problem: reducing the massive computational and environmental costs of training frontier-scale models from scratch.
- **Originality:** While model expansion is an established concept, the paper provides a novel geometric insight by arguing that the “trap” is not merely an initialization failure, but a byproduct of how standard element-wise optimizers interact with function-preserving landscapes.

## Weaknesses

- **Generalization vs. Scope:** The authors frame their findings as a general geometric pathology inherent to width expansion, yet the empirical validation is limited to decoder-only Transformers. It remains unclear whether these findings generalize to other architectures especially since earlier methods like Net2Net demonstrated successful width expansion in computer vision models without encountering similar stagnation.
- **Rigidity of Theoretical Assumptions:** The theoretical framework relies on specific assumptions (e.g., the loss landscape being an analytic function of the parameters). The paper does not sufficiently clarify whether the method fails if these assumptions are violated ; for instance, when using non-analytic activations like GELU or whether there are architectural edge cases where the “trap” naturally does not form.
- **Optimization State and Hyperparameters:** The analysis does not fully decouple the “Subspace Trap” from standard optimization challenges. It is unclear whether the observed stagnation is a fundamental geometric constraint or if it could be mitigated through more aggressive tuning of learning rates or resetting optimization states for newly added parameters. Furthermore, the paper assumes the same training hyperparameters for both scratch and expanded models, which may not be optimal for warm-started regimes.

---

> ### Author Rebuttal · Authors · 2026-03-31
>
> We thank the reviewer for the thorough and balanced assessment, and for recognizing the soundness, presentation quality, and significance of the work.
> > Q1: Generality Beyond LLMs
>
> Our work builds on the decoder-only Transformer expansion literature (SCALE, HyperCloning) and our primary experiments are conducted in this setting. To address the reviewer's concern about generality, we additionally conducted a ViT width expansion experiment. We expanded a pretrained ViT-Small (d=384) → ViT-Base (d=768) on ImageNet-1K with symmetric expansion, training for 1500steps:
>
> |   |   |   |
> |---|---|---|
> |Model|Val acc|Block Similarity|
> |ViT-Small|0.7473|0.4444|
> |Expanded ViT-Base （AdamW）|0.8159|0.9351|
> |Expanded ViT-Base （Muon）|0.8180|0.9180|
>
> The Subspace Trap phenomenon clearly transfers to ViTs: expanded models exhibit similarity > 0.91, far above the seed model's natural level (0.4444). Muon shows a lower similarity than AdamW (0.9180 vs 0.9351), consistent in direction with our LLM findings. The relatively small gap is expected given the limited training budget (1500 steps); in our LLM experiments, the Muon–AdamW divergence also widens progressively with training (Fig. 4).
>
> > Q2: Sensitivity to Optimization Parameters
>
> ****LR Ablation**** on SmolLM2-0.54B with Symmetric Expansion:
>
> |   |   |   |
> |---|---|---|
> |Peak LR|Optimizer|Block Similarity after training 30B tokens|
> |5e-4|AdamW|0.923|
> |5e-4|Muon|0.867|
> |1e-3 (2×)|AdamW|0.896|
> |1e-3 (2×)|Muon|0.797|
>
> Doubling LR only marginally reduces AdamW's similarity (0.923→0.896), still far above Muon at default LR (0.867). Even at 2× LR, AdamW cannot match Muon's escape — the gap is directional, not magnitude-based. AdamW's element-wise scaling preserves block correlations regardless of step size; larger steps move further in the *_same_* confined subspace.
>
> Regarding warmup and state reset:
>
> - We expand from open-source pretrained checkpoints, which do not release optimizer states — training necessarily starts fresh.
>
> - Warmup controls the magnitude ramp-up but not the update direction. Neither can alter AdamW's element-wise structure that preserves block correlations. The trap is geometric, not a tuning problem.
>
> > Q3: When Does The Assumptions Fail
>
> The River-Valley loss landscape has been extensively studied in the literature. Our theoretical framework relies on assumptions commonly associated with this landscape: (1) an analytic loss function, (2) a bounded Hessian, (3) the existence of an Eigen Gap, (4) the uniqueness and continuity of the subspace, and (5) a small learning rate.
>
> The first three are widely accepted assumptions utilized across numerous theoretical works, while assumption (5) is a standard experimental setting. Regarding assumption (4) and the overall applicability of the theory, existing literature provides strong support for its relevance in practical regimes. Specifically, empirical results from [1] demonstrate that the River-Valley assumption holds for a wide variety of architectures—including VGG16, ResNet18, ViT, and GPT-2—spanning multiple modalities and tasks, such as Transformers and language modeling. Additionally, [2] provides a constructive proof justifying the application of the River-Valley framework to FFN models.
>
> At present, it remains unclear exactly under what specific conditions or edge cases this assumption might explicitly fail. However, we respectfully maintain that this does not weaken the core theoretical contributions of our paper. Our analysis still provides a rigorous and valuable contribution toward fundamentally understanding the relationship between optimizer symmetry and model width expansion.
>
> [1] Y. Zhang et al., "Why transformers need adam: A hessian perspective." NeurIPS, 2024.
>
> [2] K. Wen et al., "Understanding warmup-stable-decay learning rates: A river valley loss landscape perspective." ICLR, 2025.

---

> > ### Author Rebuttal · Reviewer_v95r · 2026-04-03
> >
> > Thank the authors for the response. I will keep my score for time being

---

> > > ### Author Response · Authors · 2026-04-07
> > >
> > > We thank the reviewer for the time and thoughtful feedback throughout the discussion. We hope our responses have addressed the concerns raised.

---

### Official Review · Reviewer_VtCi · 2026-03-10

**Soundness:** 2
**Presentation:** 3
**Significance:** 3
**Originality:** 4
**Overall Recommendation:** 4
**Confidence:** 3

**Summary:**

In model width expansion, where a trained smaller model's weights are used as sub-blocks of a larger model at initialization, this work found a phenomenon called subspace trap: the sub-blocks of the large model remain high similarities with each other, and the weights are effectively in a subspace without fully exploiting the larger width. To escape from the subspace, Muon optimizer is preferred as it updates different directions in a more isotropic manner. Finally, escaping subspace traps and activating new dimensions are verified to have a performance gain.

**Compliance With Llm Reviewing Policy:**

Affirmed.

**Final Justification:**

This paper shows very interesting empirical results and potentially can inspire a series of future studies. However, their theoretical explanation is not convincing to me. I outweight empirical results over theory, and therefore suggest weak acceptance despite the non-convincing theory.

**Key Questions For Authors:**

Questions affecting my evaluation:
1. If the river valley picture is correct, after width expansion, the river directions and valley directions remain there. Escaping from the subspace is likely due to the directions orthogonal to the subspace rather than the valley directions. The key is to study the orthogonal directions. If the river valley picture is incorrect, let's say the small model is trying to reach its global minimum, then the expanded large model at initialization is likely close to some local minimum or saddle point, as the small model is trained well. To escape from the local minimum, the key is still making use of the directions orthogonal to the subspace. The key question is about the new directions in the large model, regardless of whether there are river valleys. My question is whether the new directions pull the model out of the subspace, but with very small gradients (like escaping from saddle points), or push the model into the subspace (like escaping from a local minimum). You can study the gradients or Hessian in different subspaces.
2. What determines the difficulty of escaping the subspace? One point I observe is how symmetric the expansion is. Based on Figures 2 and 5, I can conclude that Symmetric Expansion makes escaping more difficult than Diagonal Expansion. Intuitively, Symmetric Expansion requires 4 blocks to escape from the trap together, which can be more difficult.
3. Based on 2, I feel there may exist a better way of expanding, i.e., copying $W_s$ to the upper-left block only and making all other blocks zero. This expansion preserves functions and has the least symmetry. Why didn't people explore this?
4. Figure 4 is a little funny; Muon still has very high similarity.
5. What is the similarity between blocks of models trained from scratch? We need this as a baseline. If the model trained from scratch can reach a very low similarity, it seems to have a similar final loss value to that from expanded models, which have high similarity (~0.9). What will this mean?

Questions out of curiosity:

6. How will your results tell us about neural scaling laws, especially the width scaling? And can you use current neural scaling laws to estimate the desired loss after width expansion? Can you estimate the compute saved to reach the optimal-compute frontier by doing width expansion rather than training from scratch? Do you expect the width expansion to beat the current neural scaling laws, which are only measured from models trained from scratch?

**Limitations:**

The paper has not discussed limitations.

I think the empirical results are interesting, but the theoretical explanations are not convincing or complete.

It may even be feasible to soften the assumptions and arguments. A too specific picture, like the river valley, yet not strongly supported by data nor predictive, can lead to unnecessary criticisms.

**Strengths And Weaknesses:**

Strengths:
1. The discovery of subspace traps in LLMs is interesting and novel.
2. Experiments are systematic and convincing.

Weaknesses:
1. The theoretical analysis is less convincing. Whether or not the landscape is a river valley, the subspace should persist (see key question 1). The river valley picture also could not predict the difficulty of escaping. The empirical results do not reject the river valley picture, but also do not strongly support it.

---

> ### Author Rebuttal · Authors · 2026-03-31
>
> We sincerely appreciate your time and effort in reviewing our paper and providing valuable comments.
> > W1 & Q1: Relation between optimizer geometry and River-Valley analysis
>
> The river valley provides valuable insight for understanding MWE with geometry-aware optimizers. Its core observation is that flat directions (rivers) between valleys contain optimization opportunities, lower loss exists along these directions, but their near-zero Hessian eigenvalues make it difficult to optimize. Our Proposition 3.3 shows that width expansion inherits and even amplifies this difficulty (depending on the expansion matrix $P$), leaving the expanded subspace underutilized. Introducing a geometry-aware optimizer (Muon) serves a dual purpose: it ensures progress along the river while maintaining convergence in the valley. If river-valley doesn't hold as the reviewer say, escaping a local minimum might leads to discarding the seed model's knowledge, where expansion could not guarantee improvement over random initialization.
>
> Prior work [1] empirically validates the Hessian eigenvalue distribution of Transformers (their Fig. 1), confirming the coexistence of strongly convex subspaces (valleys) and flat subspaces (rivers). Our Fig. 2 supports this structure. Thus, our analysis provides a valuable theoretical understanding with empirical support.
>
> [1] Y. Zhang et al., "Why Transformers Need Adam: A Hessian Perspective" NeurIPS, 2024.
>
> > Q2: The difficulty of escaping the subspace
>
> The escape difficulty is related to the Hessian structure in the expanded subspace, which depends on the eigenvalues of the expansion matrix $P$. Symmetric expansion ($P_1$, eigenvalues ${0, 1}$) and diagonal expansion ($P_2 $, eigenvalues ${1, 1}$) produce different landscape geometries, and our experiments confirm that the two strategies yield different $\mathcal{S}_{block}$ trajectories under the same optimizer. Notably, the core finding — that element-wise optimizers fail to escape while geometry-aware optimizers succeed — holds consistently across both expansion types (Figs. 2–4). A finer-grained theoretical comparison between expansion strategies is an interesting direction for future work.
>
> > Q3: Upper-left-block-only expansion
>
> Placing $W_S$ only in the upper-left block is a natural idea to avoid tiled redundancy. However, as analyzed in LEMON [2], it faces a fundamental tension with RMSNorm: without noise, $W_L = [W_S, 0; 0, 0]$ is function-preserving but new parameters receive zero gradient (cold-start); with noise, the RMS ratio becomes data-dependent and no static $\gamma$ maintains equivalence (losslessness is lost). Symmetric/Diagonal expansion avoids this by placing $W_S$ in all diagonal blocks, keeping the $[h; h]$ pattern self-consistent with RMSNorm.
>
> [2] Y. Wang et al., "LEMON: Lossless model expansion." ICLR, 2024.
>
> > Q4&Q5: Similarity baseline for models trained from scratch
>
> The seemingly "high" similarity in Figure 4 highlights the dual nature of $\mathcal{S}_{block}$: its absolute value reflects retention of seed knowledge, while its trajectory (drop from 1.0) reflects activation of new capacity. Both aspects matter. We measured the similarity for models trained from scratch and from expansion (layer 24, up_proj, Qwen3-2.38B):
>
> | Model | $\mathcal{S}_{block}$ |
> |-------|----------------------|
> | Scratch-trained (AdamW) | 0.475 |
> | Scratch-trained (Muon) | 0.495 |
> | Expanded + AdamW | 0.880 |
> | Expanded + Muon | 0.788 |
>
> The natural baseline is ~0.48. Expansion initializes near 1.0; AdamW stays trapped at 0.880 (nearly 2× baseline), while Muon reduces it to 0.788. Staying above 0.48 is desirable (preserving seed knowledge), but a reduction from 1.0 is necessary to utilize added parameters. SVD pruning (Fig. 6) confirms this: pruning tail components causes negligible degradation for AdamW (dead redundancy) but severe degradation for Muon (new knowledge encoded). Muon converts structural redundancy into effective capacity purely through optimizer geometry.
>
> > Q6: Scaling laws and compute savings
>
> Current scaling laws [3][4] assume from-scratch training and do not account for expansion. Our results suggest width scaling efficiency is optimizer-dependent under expansion: AdamW-expanded models underperform their parameter count — updates remain confined to the seed subspace, effectively behaving as over-parameterized small models. **Muon-expanded models escape this trap and achieve lower loss than both AdamW-expanded and scratch-trained models at matched token budgets (Fig. 5), with 32–44% fewer tokens to reach AdamW's final loss.** This suggests **width scaling with appropriate optimizers can exceed the from-scratch scaling frontier**, though precisely quantifying this requires fitting scaling laws on our training distribution — we leave this to future work.
>
> [3] J. Kaplan et al., "Scaling Laws for Neural Language Models," arXiv, 2020.
>
> [4] J. Hoffmann et al., "Training Compute-Optimal Large Language Models," arXiv, 2022.

---

> > ### Author Rebuttal · Reviewer_VtCi · 2026-04-02
> >
> > I thank the authors for their effort.
> >
> > **Q1 and W1** I am still not convinced (which can be my problem). The high-level argument I have is that the trap is due to the fact that the small model is already trained very well, which is irrelevant to whether the landscape is a river valley or not. I personally also believe it is a river valley. To really address this issue, one needs to do a mechanistic interpretation type study: find the directions that the model is following to escape the trap, and understand the meaning of those directions empirically.
> >
> > **Q2** I look forward to future work on this!
> >
> > **Q3** Thank you for the information.
> >
> > **Q4 and Q5** Thank you for the additional analysis. It is very interesting that "Expanded" always has much higher similarity than "Scratch". But their loss value can be similar, right? How should I interpret this fact? And if the authors' picture is that lower similarity is related to better performance (escaped from the trap), it will indicate that training from scratch is always better. This is in contradiction to your answer to my Q6.
> >
> > To summarize, I think the empirical results are very interesting and potentially can lead to more insights. This alone makes me want to accept the manuscript. However, the analysis provided cannot convince me, and I therefore gave weak acceptance. At this stage, if the author can address my new questions above related to **Q4 and Q5**, I am happy to raise my score to 5.

---

> > > ### Author Response · Authors · 2026-04-07
> > >
> > > We sincerely thank the reviewer for the continued engagement and constructive feedback during the discussion period!
> > >
> > > **Q1&W1:** We appreciate the reviewer's suggestion. A mechanistic interpretability study to identify the specific escape directions is a valuable direction. Due to the complexity of such analysis, we leave this for future work.
> > >
> > > **Q4&Q5:** We thank the reviewer for this important follow-up. We first re-emphasize that our setting is **efficient training** — the goal of width expansion is to reach target loss with fewer tokens by inheriting seed model knowledge. Under this setting, **$\mathcal{S}_{block}$ is not a performance metric** — it measures the structural redundancy between expanded blocks and seed blocks, i.e., how much the expanded model still resembles its initialization.
> > >
> > > Crucially, our claim applies **only to comparing expanded models against each other** (e.g., AdamW vs. Muon), not to comparing Expanded vs. Scratch: for expanded models, a reduction from the function-preserving initialization (near 1.0) is necessary to activate the added width. It does not imply that lower similarity is universally better across different training settings. Indeed, **Expanded models achieve lower loss than Scratch at matched token budgets despite higher similarity**, as shown below:
> > >
> > > | Settings | Target Loss | AdamW FLOPs (1e20) | Muon FLOPs (1e20) | Muon Savings |
> > > | -------------------------------- | ----------- | ------------------ | ----------------- | ------------ |
> > > | Qwen-train-from-scratch | 1.793 | 9.498 | 7.313 | 23.01% |
> > > | Qwen-symmetric width expansion | 1.648 | 9.030 | 5.782 | 35.97% |
> > > | SmolLM-train-from-scratch | 1.859 | 0.966 | 0.835 | 13.57% |
> > > | SmolLM-symmetric width expansion | 1.768 | 0.966 | 0.535 | 44.61% |
> > >
> > > This directly shows that preserving seed knowledge is beneficial and that the goal is not to match Scratch-level similarity.
> > >
> > > To validate whether the similarity drop reflects **useful capacity activation**, we further conduct SVD pruning experiments in paper: Expanded+AdamW is robust to pruning → dead redundancy; Expanded+Muon is sensitive to pruning → new knowledge encoded. Thus Muon achieves the desirable regime: a **controlled drop** that retains seed knowledge while activating new capacity.
> > >
> > > In summary, $\mathcal{S}_{block}$ is a diagnostic, not an objective. Scratch has low similarity because it has no seed to preserve. Expanded models should stay more similar than scratch, but they must reduce similarity enough to make the added width functional. AdamW only partially achieves this, while Muon does so much more effectively.
> > >
> > > We hope our clarifications have addressed the reviewer's concerns.

---

### Official Review · Reviewer_ZJ1g · 2026-03-15

**Soundness:** 3
**Presentation:** 3
**Significance:** 3
**Originality:** 3
**Overall Recommendation:** 4
**Confidence:** 1

**Summary:**

This work focuses on an important question of scaling an existing model by increasing its width, and argues that if done naively, the model will be stuck in a region of limited expressivity. It also proposes new methods to escape such suboptimal regions.

**Compliance With Llm Reviewing Policy:**

Affirmed.

**Final Justification:**

Thanks to the authors for the response! I maintain my original rating.

**Key Questions For Authors:**

- Around Line 49 (Right-hand side): "...with preserved inference latency..." Is this true? If you expand the width, why is the latency preserved?
- Figure 1: What would the similarity be for a randomly initialized model?

**Limitations:**

Limitations are not explicitly discussed in a separate section, but societal impact is discussed.

**Strengths And Weaknesses:**

The reviewer wanted to preface the assessment by acknowledging a lack of domain expertise in this area, and hence, the comments below should be treated with a grain of salt. A confidence of 1 was used as well.

In terms of strength.
- The authors provided a theoretical analysis to understand the empirical results.
- The authors conducted well-planned and well-executed empirical experiments to examine whether the analysis holds in practice.
- Care is taken to make sure the experimental results reflect the analysis, not just optimizer differences.
- Writing (other than the theory part) is easy to follow, even for readers who aren't familiar with this area (myself).
- The proposed analysis is novel.

Despite the authors' considerations, I find it hard to determine how much the quality differences are caused by optimizer variations. I appreciate their effort in conducting pretraining-from-scratch experiments, but it remains unclear whether the differences between pretraining and continual pretraining can be directly compared or combined.

---

> ### Author Rebuttal · Authors · 2026-03-31
>
> We thank the reviewer for the careful and honest assessment. We address each concern below.
> > Q1: Disentangling optimizer variations from quality differences
>
> We appreciate the reviewer's careful attention to this confounding factor. To isolate the expansion-specific effect from the general optimizer advantage, we employ a difference-in-differences design. Concretely, for each setting we take the final loss achieved by AdamW as the target and measure the FLOPs each optimizer requires to reach it, for Qwen3-2.38B (expanded from Qwen3-0.6B) and SmolLM2-0.54B (expanded from SmolLM2-135M):
>
> | Settings                         | AdamW FLOPs (1e20) | Muon FLOPs (1e20) | Muon Savings |
> | -------------------------------- | ------------------ | ----------------- | ------------ |
> | Qwen-train-from-scratch          | 9.498              | 7.313             | 23.01%       |
> | Qwen-symmetric width expansion   | 9.030              | 5.782             | 35.97%       |
> | SmolLM-train-from-scratch        | 0.966              | 0.835             | 13.57%       |
> | SmolLM-symmetric width expansion | 0.966              | 0.535             | 44.61%       |
>
> If the gap were purely "Muon is a better optimizer," the savings ratio would remain constant across settings. Instead, **Muon's advantage increases substantially after expansion: 23%→36% for Qwen and 14%→45% for SmolLM**. These additional savings specifically quantify the benefit of escaping the Subspace Trap — capacity that remains locked in the seed subspace under AdamW. The effect is consistent across two model families of different scales.
>
> > Q2: Clarification on preserved inference latency with width expansion
>
> We thank the reviewer for flagging this imprecise language. What we intended to convey is that width expansion preserves the sequential depth (i.e., the number of layers), which is the dominant factor governing autoregressive inference latency, given the inherently sequential token-by-token generation process. **Wider matrices increase per-step FLOPs, which are efficiently parallelized on modern GPUs/TPUs, far less than depth expansion, which adds sequential steps.** We will revise to “with substantially lower inference latency overhead compared to depth expansion.”
>
> > Q3: Similarity metric for a randomly initialized model
>
> We computed the block similarity matrix for randomly initialized models trained from scratch under identical settings (70B/30B tokens).  Block similarities range from 0.46 to 0.61 — far below the >0.90 observed in expanded models (Fig. 1). This confirms that the high block redundancy is not a natural consequence of architecture or data, but a pathology unique to function-preserving expansion. The scratch-trained similarity matrices are provided at https://anonymous.4open.science/r/rebuttal-D8B8/comparison_qwen-smollm_adam_scratch.png for direct comparison with Fig. 1.
>
> > Limitations section.
>
> We will add an explicit limitations section in the revision discussing: (1) the current focus on decoder-only Transformers, and (2) the need for broader architectural validation.

---

> > ### Author Rebuttal · Reviewer_ZJ1g · 2026-04-03
> >
> > Thanks to the authors for the response! I maintain my original rating.
> >
> > This is not important, but I'm not convinced by the answer that increasing width is better than depth from an inference latency perspective. For most inference setups, it's memory-bound (i.e., the FLOPs don't matter as much as training, but the amount of on-off-chip memory access does). I could be wrong or missing something, but I'd like to see some actual latency measurements if the authors want to be sure about it. However, for the purposes of this rebuttal, this is not a big deal for me.

---

> > > ### Author Response · Authors · 2026-04-07
> > >
> > > We thank the reviewer for the insightful remark. We agree that inference is often memory-bound and acknowledge that actual latency measurements would strengthen the argument. Recent work [1] empirically demonstrates that, under the same parameter budget, shallower-wider models achieve lower latency than deeper-thinner ones, as depth increases sequential operations while width leverages GPU parallelism.
> > >
> > > [1] Y. Fu et al., "Nemotron-Flash: Towards Latency-Optimal Hybrid Small Language Models" NeurIPS, 2025.

---

### Decision · Program_Chairs · 2026-04-30

**Decision:**

Accept (regular)

**Comment:**

The reviewers are unanimous in recommending acceptance. The identification of the subspace trap is an interesting empirical finding and the evidence for the muon-based solution is clear. The paper should provide useful work for potentially making model width expansion more practical.